# Prioritizing Disease Diagnosis in Neonatal Cohorts through Multivariate Survival Analysis: A Nonparametric Bayesian Approach

**DOI:** 10.3390/healthcare12090939

**Published:** 2024-05-02

**Authors:** Jangwon Seo, Junhee Seok, Yoojoong Kim

**Affiliations:** 1School of Electrical Engineering, Korea University, Seoul 02841, Republic of Korea; jwein307@korea.ac.kr (J.S.); jseok14@korea.ac.kr (J.S.); 2School of Computer Science and Information Engineering, The Catholic University of Korea, Bucheon 14662, Republic of Korea

**Keywords:** precedence analysis, multivariate survival analysis, nonparametric Bayesian, neonatal, disease diagnosis

## Abstract

Understanding the intricate relationships between diseases is critical for both prevention and recovery. However, there is a lack of suitable methodologies for exploring the precedence relationships within multiple censored time-to-event data, resulting in decreased analytical accuracy. This study introduces the Censored Event Precedence Analysis (CEPA), which is a nonparametric Bayesian approach suitable for understanding the precedence relationships in censored multivariate events. CEPA aims to analyze the precedence relationships between events to predict subsequent occurrences effectively. We applied CEPA to neonatal data from the National Health Insurance Service, identifying the precedence relationships among the seven most commonly diagnosed diseases categorized by the International Classification of Diseases. This analysis revealed a typical diagnostic sequence, starting with respiratory diseases, followed by skin, infectious, digestive, ear, eye, and injury-related diseases. Furthermore, simulation studies were conducted to demonstrate CEPA suitability for censored multivariate datasets compared to traditional models. The performance accuracy reached 76% for uniform distribution and 65% for exponential distribution, showing superior performance in all four tested environments. Therefore, the statistical approach based on CEPA enhances our understanding of disease interrelationships beyond competitive methodologies. By identifying disease precedence with CEPA, we can preempt subsequent disease occurrences and propose a healthcare system based on these relationships.

## 1. Introduction

In the realm of medical research, understanding the multifaceted relationships between diseases is pivotal for both prevention and recovery. Current investigations leverage probabilistic models to decode interactions between diseases and symptoms and employ genetic analyses to construct disease networks [1,2,3,4,5,6]. Additionally, the dynamics between viruses, notably those responsible for the common cold and flu, are scrutinized to unravel epidemiological trends [7]. Such correlations, like the well-documented link between respiratory infections and otitis media, underscore the importance of identifying disease connections [8,9]. These are not mere statistical ventures but clinical imperatives that guide the development of preventive and therapeutic strategies.

This study focuses on the precedence relationships among diseases, which is an aspect continually explored in medical science. Active research into the temporal relationships between diseases or between diseases and health states is shedding light on how diseases can emerge and influence one another [10,11,12,13,14]. Therefore, understanding the precedence relationships among diseases not only enhances our grasp of disease epidemiology but also significantly impacts medical practices, including the development of prevention and treatment protocols.

In this study, we applied the survival analysis theory to statistically analyze the relationships between diseases [15,16,17]. Survival analysis is a statistical analysis and forecasting technique that considers the probability of an event along with the variable of time [18,19,20,21]. Univariate and multivariate inference methods are employed for survival analyses. Univariate survival analysis is the analysis of a dependent variable, and multivariate survival analysis is applied when an event occurs for several variables.

Current research grounded in clinical outcomes predominantly involves multivariate analysis, where unraveling the intricate relationships between events stands as a significant challenge in the field of biology. However, most clinical outcome datasets consist of censored data, complicating the application of such analyses to understand the relationships between clinical outcomes. Consequently, although various methods have been proposed to handle censored datasets, analyzing censored multivariate event data remains challenging [22,23]. The analysis of such data necessitates considering the complex interrelations among multiple variables, which can be efficiently explored using nonparametric multivariate approaches [24,25]. These efforts have provided unprecedented opportunities to systematically study the relationships between diseases.

This paper introduces the Censored Event Precedence Analysis (CEPA), which is a statistical approach that calculates the precedence probability of events in multivariate datasets inclusive of censored events based on nonparametric methodology. CEPA demonstrates enhanced performance in datasets containing censored events, offering a clearer understanding of the sequence in which diseases occur. When applied to data from the National Health Insurance Service, containing a significant amount of censored information, this methodology identified the precedence among diseases. This enables the prediction of subsequent disease occurrences following an initial diagnosis, providing a basis for preemptive measures against potential future diseases.

Our discoveries suggest a link between the order in which diseases are diagnosed and the occurrence of multiple diseases at the same time, enhancing our understanding of how diseases are connected. By integrating the strengths of statistical analysis with clinical knowledge, we are forging a more robust approach to patient care. This opens up new paths for research and the development of better treatment methods. By bringing together clinical expertise and sophisticated analysis methods, this research highlights the value of working across different fields to improve health outcomes. This sets the foundation for future studies that can be both clinically relevant and based on solid statistical evidence, making complex ideas more accessible.

Our contributions in this paper can be summarized as follows:We propose the CEPA approach, a statistical method that demonstrates improved performance over existing methods for analyzing censored multivariate event datasets.Through CEPA, we enable the analysis of precedence relationships among censored events. In this study, we apply it to the National Health Insurance Service dataset to derive precedence among diseases.CEPA allows for the identification of associations between occurrences of diseases, enhancing our understanding of their interactions.

## 2. Material and Methods

### 2.1. Data Sources

In this study, we utilized a publicly accessible dataset that was free from any licensing constraints. The primary data source was the diagnosis history from the National Health Insurance Service, archived in the repository of the Ministry of the Interior and Safety, Republic of Korea (https://www.data.go.kr/en/data/15007115/fileData.do, accessed on 10 March 2023) [26]. This dataset consists of anonymized clinical details of patients, systematically categorized according to disease codes, and was applied to the study. The dataset utilized in this research encompassed 1,089,605 patient IDs, ages, 2231 disease codes, and the times that each patient was diagnosed with diseases corresponding to these codes.

The disease classification for this study adheres to the International Classification of Diseases (10th Revision) (ICD-10), which is a system developed and published by the World Health Organization (WHO) [27]. This system uses a combination of letters and numbers to categorize diseases and health-related conditions, with each letter representing a specific category of diseases or conditions. Appendix A provides the disease codes, incidence rates among neonates, and descriptions for the 27 disease categories classified according to the ICD-10 major classification. The ‘D’ and ‘H’ disease groups are further subdivided based on the ICD-10 disease categories and specific conditions. More detailed information on this classification can be found on the WHO’s official website (https://icd.who.int/browse10/2019/en, accessed on 10 March 2023).

To avoid left-censoring data, our study focused on neonatal data. The dataset, derived from the National Health Insurance Service of South Korea, spans from 2002 to 2016 and comprises a random selection of one million citizens [26]. Given that 2002 had the highest birth rate within our data timeframe, it was chosen as the experimental year, focusing on 9565 newborns.

Our research selected the top seven disease categories with the highest incidence rates among neonates from groups named under the same disease description in Appendix A, which were classified according to the ICD-10 major categories. Diseases with lower diagnosis rates were excluded from the analysis due to difficulties associated with studying their precedence and disease network analysis, thereby focusing our research on those with a diagnosis rate of over 70%. Along with the top seven diseases, for analytical convenience, the disease groups in the ICD-10 are categorized as infection, eye, ear, respiratory, digestive, skin, and injury diseases based on their descriptions. The ICD-10-based disease groups, their names, and the number of diagnosed cases and diagnosis rates can be found in Table 1. Patients diagnosed with more than two diseases within a year were then selected, yielding data for 9533 patients. The date of the first onset was defined as the earliest occurrence of a similar disease. A total of 4343 individuals constituted the group that experienced all seven diseases with a high incidence rate. A flowchart of the data collection and organization process is shown in Figure 1. Building on this data foundation, we analyzed the relationships between various censored events, employing advanced statistical methodologies, as detailed below.

### 2.2. Applying Methodology for Joint Probability Density Estimation

The proposed CEPA method identifies significant correlations based on the two-time-to-event data within datasets containing censored occurrences, thereby determining precedence. The CEPA utilizes a nonparametric estimation approach based on the optional Pòlya tree (OPT) to estimate the joint PDF, which is crucial for analyzing precedence in event data. This method can be used for handling censored time-to-event data in survival data analysis [28]. Before discussing the methodology, it is necessary to first define the variables as follows: for sample i, $T_{1}$ and $T_{2}$ indicate the occurrence times of two events, whereas $C_{1}$ and $C_{2}$ represent the censoring times for these events, respectively. An event is considered censored if its censoring time precedes its occurrence time. In such cases, the data are represented as $X_{i1}$ and $X_{i2}$ instead of $T_{i1}$ and $T_{i2}$, where $X_{i1}=\min\left(T_{i1},C_{i1}\right)$ and $X_{i2}=\min\left(T_{i2},C_{i2}\right)$.

The OPT determines the likelihood $\Phi \left(A\right)$ across regions in a recursively partitioned sample space Ω by dividing a region A into sub-regions and calculating their likelihoods. When a region A is divided along the $T_{i}$ axis into sub-regions $A_{ij}$, the likelihood function simplifies as follows:(1)$$\Phi \left(A\right)=\frac{1}{2}\Phi_{0}\left(A\right)+\frac{1}{4}\overset{2}{\underset{i=1}{\sum }}B^{\prime }\left(N\left(A_{i1}\right),N\left(A_{i2}\right)\right)\Phi \left(A_{i1}\right)\Phi \left(A_{i2}\right),$$
where $\Phi_{0}\left(A\right)$ represents the likelihood for samples uniformly distributed within A, and $B^{\prime }$ simplifies the adjusted Beta function ratio, with $B^{\prime }\left(x,y\right)$ matching the ratio $\frac{B\left(x+0.5,\;y+0.5\right)}{B\left(0.5,0.5\right)}$. $N\left(A\right)$ indicates the count of samples within A and the OPT seeks a uniform distribution across partitions, with the sample count defining the density of each.

To accommodate censored data, which cannot be directly counted within A, sample numbers are inferred via the joint distribution $f\left(T_{1},\;T_{2}\right)$, denoted as N(A|f), as follows:(2)$$f\left(T_{1},\;T_{2}\right)=\;\mathrm{OPT}\left(N\left(A|f\right)\right),$$
enhancing the analysis of censored observations in survival data studies.

### 2.3. Censored Event Precedence Analysis

Using traditional OPT analysis to estimate precedence in scenarios with censored data turns out to be challenging. Therefore, this research introduces CEPA, an analytical methodology for precedence among censored events derived from OPT, capable of calculating the PDF in censored multivariate data, as demonstrated in Equation (2). CEPA is designed to estimate the precedence relationships between events, incorporating censored data by evaluating the conditional probability values between two events. A significant difference in these conditional probabilities signifies the potential for one event to precede another.

CEPA estimates precedence relationships by calculating the joint probability of bivariate events in censored datasets. However, extending beyond bivariate analysis to multivariate data leads to practical limitations in terms of computational capacity and sample size. Therefore, for multivariate data, rather than applying CEPA directly, the approach constructs sequences through combinations of all pairs in bivariate datasets, calculating the likelihood of these sequences. The likelihood $L\left(\cdot \right)$ of a sequence involving n time-to-event data $\left[T_{1},T_{2},T_{3},\cdots ,T_{n}\right]$ can be expressed as follows:(3)$$L\left(T_{1}\to T_{2}\to T_{3}\to \cdots \to T_{n}\right)=\;\mathrm{Pr}\left[T_{2}\left|T_{1}\right.\right]\mathrm{Pr}\left[T_{3}\left|T_{1},T_{2}\right.]\cdots \mathrm{Pr}[T_{n}\left|T_{1},T_{2},\right.\cdots ,T_{n-1}\right].$$

Here, $T_{1}\to T_{2}$ represents the sequence where event 1 occurs first, followed by event 2. Based on the methodology of Equation (3), the likelihood of a comparison of sequences in multivariate data facilitates the estimation of precedence relationships.

### 2.4. Multivariate Survival Analysis with CEPA

The CEPA methodology presented in this research is an approach for inferring precedence analysis within the analysis of correlations among multivariate data. We conducted comparisons of the median time-to-event and determined the likelihood of sequences composed of multivariate data using CEPA, facilitating the inference of precedence relationships among events.

The median time-to-event represents the time by which half of the sample experienced the event. In a dataset composed of time-to-event data, events not occurring within the maximum observation period are considered censored. Accordingly, we define the median considering censored data as the overall median and the median without considering censored data as the observed median. The overall median time includes the maximum observation day, while the observed median time encompasses events estimated to have occurred within the maximum observation period. For event A, represented as $T_{A}$, in univariate analysis, the overall median time to the event is calculated as the latest time when the marginal survival probability exceeds 0.5, as follows:(4)$$median_{overall}\left(T_{A}\right)=\sup\left\{t:\mathrm{Pr}\left[T_{A}\geq t\right]\geq 0.5\right\}.$$

For all events, T, the observed median time accounts only for cases where events occur within the maximum observation time, as follows:(5)$$median_{observed}\left(T_{A}\right)=\sup\left\{t:\mathrm{Pr}\left[T_{A}\geq t|T_{A}\leq T_{max}\right]\geq 0.5\right\}.$$

The method for calculating the likelihood of sequences can be derived from Equation (3). In our study, we applied a scoring method to the likelihood of sequences, allowing for the easier comparison of likelihoods between sequences through a scored likelihood. For a dataset with n time-to-event occurrences $\left[T_{1},T_{2},T_{3},\cdots ,T_{n}\right]$, the likelihood $L\left(\cdot \right)$ can be transformed into a score as follows:(6)$$Score\left(T_{1}\to T_{2}\to T_{3}\to \cdots \to T_{n}\right)=-\log\left(L\left(T_{1}\to T_{2}\to T_{3}\to \cdots \to T_{n}\right)\right).$$

### 2.5. CEPA Simulation Setting

To demonstrate the validity and utility of the proposed CEPA method, we generated a simulation dataset with 500 samples. These samples were used to compare CEPA with a control group model. The data generation process was guided by three criteria concerning the relationships between events: (1) one event must occur before another, (2) the timing between two events should be dependent, and (3) the time interval between two events should be independent of the timing of the preceding event. We considered three events and generated three censored time-to-event data, in which $T_{1}$ preceded $T_{2}$, and $T_{2}$ preceded $T_{3}$. Each time-to-event comprised three event times ($T_{1},\;T_{2},T_{3})$. The event times were generated using uniform, lognormal, and exponential distributions, as well as the Clayton model, respectively [29]. Each censored event time comprised three censored times ($C_{1}$,$C_{2}$,$C_{3}$). Each censored time was generated from uniform, lognormal, or exponential distributions, respectively. The sample distributions are shown in Table 2. After generating 500 samples for each distribution, we compared the actual event times with the censoring points. The censored time points, $X_{i}$, were given as follows:(7)$$X_{i}=\min\left(T_{i},C_{i}\right),\;\;i=1,2,3.$$

In addition, censoring indicators, $\Delta_{i}$, were given as follows:(8)$$\Delta_{i}=I\left(T_{i}\leq C_{i}\right),\;\;i=1,2,3.$$
where $I\left(\cdot \right)$ is an indicator function. Finally, the data were preprocessed in the form of $\left\{X_{1},\Delta_{1},\;X_{2},\;\Delta_{2},\;X_{3},\;\Delta_{3}\right\}$ metrics to apply to the simulator for this study.

## 3. Results

### 3.1. Simulation Experiments and Results

Each simulation involved generating 500 data points, with the overall comparison drawn from the results of 100 simulations. Based on the simulation data generated, the performance of CEPA was compared with that of a control group model. The comparison utilized the methods developed by Dabrowska, Lin-Ying, and a simple computational model for analysis [30,31]. We adopted a naive approach using a simple computational model. This naive approach disregards censoring, focusing solely on the comparison of time-to-event data. A lower time-to-event indicates an event that occurred earlier. The method of assessing likelihood was by ordering the time-to-event data and then comparing the sequence of actual events to this order to evaluate consistency.

Figure 2 showcases the boxplot comparisons of the results across four estimation models based on different data generation distributions. From Figure 2A, it is observed that the CEPA model demonstrated the highest likelihood, 76%, for the uniform distribution. Furthermore, Figure 2B–D illustrate how CEPA outperformed the comparative models by margins of 65%, 36%, and 33% for the exponential, lognormal, and Clayton distributions, respectively. Additionally, while some models exhibited a performance that was only marginally better than the naive model, the model proposed in this study demonstrates similar or superior performance. These results highlight CEPA’s superior likelihood scores and stable performance across various distributions compared to the control groups.

### 3.2. Application Studies for Cohort

Application studies were performed using disease diagnosis time data from the National Health Insurance Cohort in Korea. We selected 9533 newborn patients from one million samples. The disease diagnosis data included infectious, ear, respiratory, digestive, skin, and injury diseases. The data period was 1 year, and preprocessing was performed to scale it from 1 to 52 so that it could be easily applied to the analysis. After scaling, each point represented 1 week in real-time. We assigned a value of 53 to the unobserved event from each patient, assuming that it was a censored event.

The censoring rates and median times of the seven disease diagnosis events for the patients are listed in Table 3. Respiratory diseases were the most frequently occurring diseases in the neonatal population, with a censoring rate of 0.126%. Digestive, ear, and injury diseases had late onset and relatively high censoring rates compared to those for other diseases. The overall patients with censored data had a longer median onset date than the observed patients without censored data. However, the censoring rate for respiratory diseases was low, resulting in similar median values. We estimated the joint probability distributions for all the possible univariate and bivariate sets of disease diagnostic events following the aforementioned example. We estimated that there were seven univariate and twenty-one bivariate joint distributions from the seven investigated disease events.

### 3.3. Univariate Survival Analysis

The univariate survival curves estimated by CEPA for the 9533 patients are plotted in Figure 3. These curves were consistent with Kaplan–Meier estimates [32,33]. According to the probability mass analysis conducted by CEPA, infectious, ear, respiratory, and skin diseases had high incidence rates in the early stages, and these rates decreased over time. However, digestive, eye, and injury diseases were evenly distributed over time. For respiratory diseases in the patient population, the probability of onset within 10 weeks was approximately 98%, with a very high initial incidence rate. The disease systems had extremely different censoring rates and median diagnosis times. As shown in Table 3 and Figure 3, respiratory diseases had an early onset and relatively low censoring rates compared with other diseases. The univariate survival curves estimated by CEPA for the observed 4343 patients are plotted in Appendix A. These curves were also consistent with the Kaplan–Meier estimates. In the patient population without censored events, the initial incidence rates of infectious, respiratory, skin, injury, and ear diseases were high but decreased over time. Eye and digestive diseases were evenly distributed over time. Respiratory diseases with low censored rates are shown in Figure 3 and Appendix A.

### 3.4. Precedence Relations for Disease Pairs

We studied the disease diagnosis precedence to determine the substantial difference in median disease diagnosis times. First, we examined the precedence between two events. For the 9533 patients, we measured the joint probability density distributions of all possible twenty-one pairs of times to the seven disease diagnosis events by CEPA (Figure 4). When comparing the joint probability densities between diseases, certain diseases had very strong precedence; for example, respiratory diseases were diagnosed before injury with a 98% chance. The diagnosis of skin, infectious, digestive, and ear diseases frequently preceded the onset of other diseases. The onset of eye diseases frequently occurred after other diseases, whereas injury followed other diseases. Interestingly, no substantial precedence was observed between the skin and infectious diseases; the same was observed for the ear and digestive diseases. Appendix A shows the bivariate joint probability density distribution for the seven diseases by applying CEPA to the 4343 observed patients. The results are similar to those shown in Figure 4, and the joint probability density difference did not exceed 1%. Among the patients diagnosed with all seven diseases, 53% were diagnosed with respiratory diseases first, and 17% were diagnosed with injury after the other diseases.

### 3.5. Precedence Analysis for Seven Disease Categories

Due to computational limitations when directly applying CEPA to multivariate datasets for precedence analysis, we calculated the likelihoods based on Equation (3) and assigned scores through Equation (6). The joint probability density distribution for the bivariate data was determined using the CEPA method introduced in this study, corresponding to the numerical values shown in Figure 4. A lower likelihood sequence score indicates a higher probability of the sequence occurring. Table 4 lists the six sequences that exhibit the most favorable likelihood sequence scores. The most probable sequence was the onset of the diseases in the order of respiratory, skin, infectious, digestive, ear, eye, and injury diseases, with a score of 6.90. The second most probable sequence was similar to the first, with only a switch for the skin and infectious diseases, with a score of 6.93. In the top six events, respiratory diseases ranked first. Based on the best sequence (score: 6.90) in Table 4, there was a probability difference of 1.37 times from a sequence with a score difference of two percent and 1.17 times from a sequence with a score difference of 1 percent. Because the difference between probabilities is large, the standard for valid sequences among many event sequences was defined as two percent. Based on the defined valid sequence range, only four-event sequences above the borderline in Table 4 were valid. The valid sequences for the patients were the onsets of respiratory diseases, followed by skin and infectious, digestive and ear, and eye and injury diseases. Appendix A shows the event sequence scores of the observed patients. There were five valid sequences for the observed patients compared with those for the overall patients. The valid sequences for the observed patients were the onsets of respiratory diseases, followed by skin and infectious, digestive, ear, eye, and injury diseases. The top six most probable sequences followed the same configuration as the sequences for the overall patients.

### 3.6. Precedence Networks

Figure 5 shows the diagnosis sequence network of the observed and overall patients based on the CEPA method results. Respiratory diseases occurred first, followed by skin and infectious diseases. The prior probabilities can be found in Figure 4 and Appendix A. Skin and infectious diseases were expressed as bidirectional diseases because the preceding probability of skin diseases was 52%, which had no practical priority. For the ear, digestive, and eye disease networks, different patient populations exhibited different results. For the overall patients, ear and digestive diseases preceded eye diseases according to the valid sequence definition. The prior probability of ear and digestive diseases was 52%, expressed in both directions. For the observed patients, eye and ear diseases exhibited bidirectionality according to the valid sequence definition. Digestive diseases preceded eye diseases, and ear and digestive diseases were expressed in both directions with a 52% prior probability. Ear diseases preceded injury because there was a sequence in which injury occurred after ear diseases in the top five valid sequences (Appendix A).

Figure 6 shows the period from the date of the respiratory disease diagnosis to the onset of infectious, eye, digestive, skin, ear, and injury diseases. Digestive, ear, and skin diseases were consistent, regardless of the time of the respiratory disease diagnosis. Within the wide range of the respiratory disease diagnosis times, spanning approximately 150 days, the digestive, ear, and skin disease onset gaps showed no strong increase or decrease from 2 to 3 weeks. Based on the consistency of the interval time, it was observed that digestive, ear, and skin diseases had a significant dependent relationship with respiratory diseases.

## 4. Discussion

The CEPA methodology introduced in this study serves as a robust framework for analyzing precedence relationships between events within censored datasets. Demonstrating superior performance over traditional methodologies like those developed by Dabrowska and Lin-Ying, CEPA validates its efficacy and reliability, particularly in contexts where data censorship significantly complicates the analysis [30,31]. Unlike naive methods that do not consider the censoring of datasets, CEPA effectively captures the censorship or complexity of event precedence. This enables a clearer analysis of the correlations between diseases. Consequently, CEPA was proven to be a more valid methodology for analyzing disease progression compared to the current methods employed by Dabrowska, Lin-Ying, and other naive approaches.

In leveraging the CEPA methodology, this study applied it to the disease diagnosis time data from the National Health Insurance cohort, notably prone to censorship issues. This study focused on the seven diseases with the highest diagnosis rates based on reclassified disease codes. The remaining codes not used in this research represented conditions in newborns with an incidence rate of less than 70%, which are insufficient for constructing a reliable disease diagnosis network.

By applying the CEPA methodology to the top seven diseases, we were able to analyze the temporal relationships between diseases effectively. This approach facilitated a comprehensive exploration of precedential disease relationships, yielding significant findings within the National Health Insurance dataset. The analysis of temporal relationships between diseases through CEPA allowed for a comprehensive exploration of antecedent disease relationships. The sequence of disease occurrence for the highest likelihood was found to be respiratory disease, followed by skin, infectious, digestive, ear, eye, and finally, injury disease. Constructing a disease sequence network structure based on these findings highlighted a clear distinction between datasets that included censored events and those that did not, showcasing CEPA’s ability to analyze precedence within censored datasets effectively. This underscores the CEPA methodology as a robust approach for analyzing censored multivariate datasets. This capability is crucial for predicting subsequent disease groups based on existing diagnoses, significantly aiding in disease prevention efforts. The adoption of this methodological approach not only enhances our understanding of disease epidemiology but also has a profound impact on medical practice by providing insights that can inform the development of targeted preventive and treatment strategies.

Furthermore, research into the correlations between diseases or health conditions has been ongoing, emphasizing the necessity of such analyses in medical research. According to a study by Heikkinen, T. and Chonmaitree, T., an understanding of the correlation between acute otitis media and respiratory viruses was developed, offering insights into reducing the incidence of acute otitis media [8]. Similarly, Ruuskanen, O. et al. have shown clear associations between acute otitis media and respiratory infections, highlighting the interconnections facilitated by respiratory viral infections [9]. Research by De Nunzio, C. et al. has explored the correlation between metabolic syndrome and prostate conditions, suggesting potential clinical implications for prevention and treatment [34]. Moreover, Nesto, R.W. analyzed the relationship between cardiovascular diseases and diabetes, detailing preventive measures in his findings, underscoring the importance of understanding these correlations [35]. Such studies persistently demonstrate the crucial role of analyzing disease correlations, which are not only imperative for further research but also provide valuable insights for disease prevention and medical practices [36,37]. However, to obtain a clearer understanding of disease correlations, it is essential to consider overlooked disease systems. Moreover, the current complex medical approaches require significant time and resources to expand and analyze overlooked disease networks. Therefore, by implementing the statistical methodology of CEPA proposed in this study to analyze disease networks, we can quickly grasp the correlations that exist among overlooked disease groups or health states. This approach integrates simple statistical methods into existing complex medical approaches, providing a comprehensive understanding of an expanded network of disease correlations. By applying the CEPA approach, it becomes possible to extend the application to a broader range of diseases than currently researched, statistically uncovering correlations among previously overlooked disease groups. This can lead to the development of more varied prevention and treatment strategies, enhancing our capability to manage health outcomes effectively.

Research focusing on the temporal relationships between diseases or between diseases and health conditions is actively progressing. For instance, Hollinger, S.K. et al. focused on uncovering the precursory conditions of amyotrophic lateral sclerosis (ALS) and exploring its correlations with other diseases [38]. Our methodology diverges from traditional medical approaches by analyzing censored time-to-event data to detect disease precedents. This approach uncovers new interpretations that have been previously overlooked. Additionally, Matthews, K.A. and Kuller, L.H. utilized cohort data to analyze the relationship between psychological risk attributes in women and metabolic syndrome [39]. Numerous studies based on cohort research have been providing valuable insights into medical practice by focusing on the temporal relationships between diseases or health states [10,11,12,13,14]. However, most of these studies are conducted with censored datasets, and many cohort studies have not adequately considered the implications of data censoring. Thus, in studies of disease correlations that rely on censored datasets, applying the CEPA methodology proposed in this study enables a more accurate analysis by taking time-to-event censoring into account. Consequently, the CEPA methodology serves as a foundation for significantly advancing research into disease relationships.

As a result, the application of CEPA to real-world data underscores the versatility of the methodology and its potential to enhance healthcare delivery. By providing a clearer picture of disease dynamics, CEPA supports healthcare professionals in making informed decisions, ultimately contributing to improved patient outcomes. Our findings, through the application of CEPA, highlight its potential to inform clinical decision-making processes by offering a deeper understanding of disease progression, which is crucial for the early detection and management of comorbid conditions. Through this comprehensive approach, the study not only addresses the technical challenges posed by data censorship but also aligns with the broader objective of advancing healthcare provision.

The CEPA methodology is anticipated to establish itself as a suitable approach within the clinical research landscape, serving as a fundamental tool for analyzing the intricate network of disease relationships. Its application in this study highlights the potential for significant advancements in understanding disease progression and in the formulation of effective healthcare strategies, marking a step forward in the quest to leverage big data for the betterment of patient care and health outcomes.

## 5. Conclusions

Understanding the intricate relationships between diseases is essential not only for targeted prevention but also for effective patient recovery strategies across the healthcare spectrum. Our study introduces a methodology that goes beyond traditional patient monitoring, aiming to anticipate and prevent subsequent diseases through a nuanced recognition of the interdependencies between diseases. The CEPA method capability to perform non-parametric estimations enables the utilization of diverse data types, such as demographic and genomic data, to provide a comprehensive analysis of the potential associations and explore preceding events within complex medical events. We hope that the proposed method can be effectively utilized by researchers and that future work will extend this approach to a broader and more detailed understanding of diseases.

## Figures and Tables

**Figure 1 healthcare-12-00939-f001:**
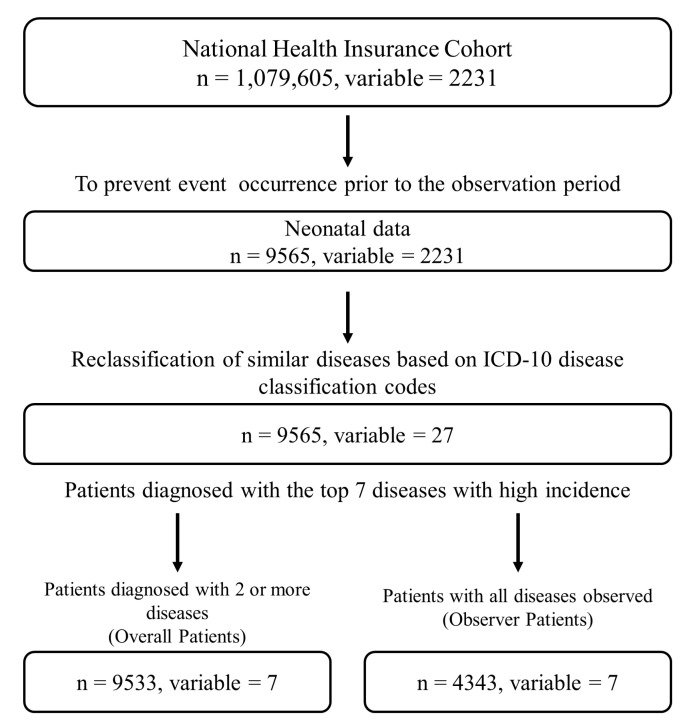
Flowchart for data collection and organization process with 9533 patients and 4343 observed patients overall.

**Figure 2 healthcare-12-00939-f002:**
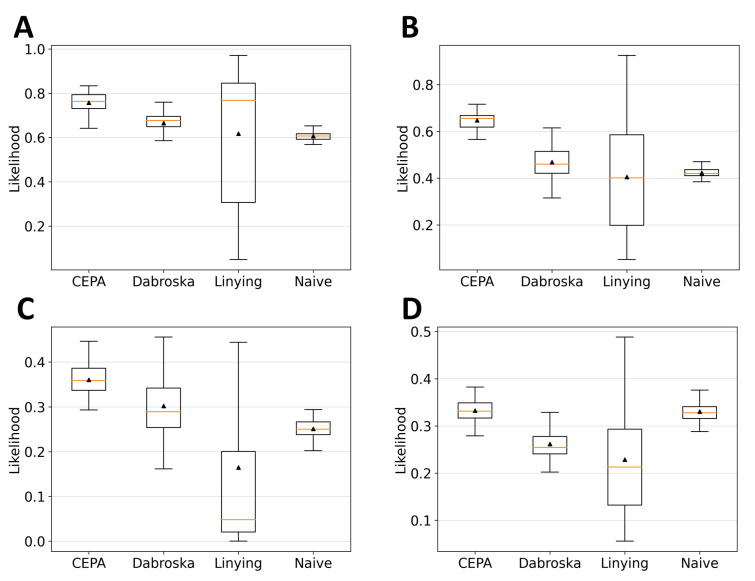
Likelihood results of the four methods (CEPA, Dabroska, Lin-Ying, and Naive models) for trivariate data generated from four distributions. (**A**) Uniform, (**B**) additive exponential, (**C**) lognormal, and (**D**) Clayton. The boxes represent the 25th and 75th percentiles, with the horizontal line inside the box indicating the median and the mean depicted by triangles.

**Figure 3 healthcare-12-00939-f003:**
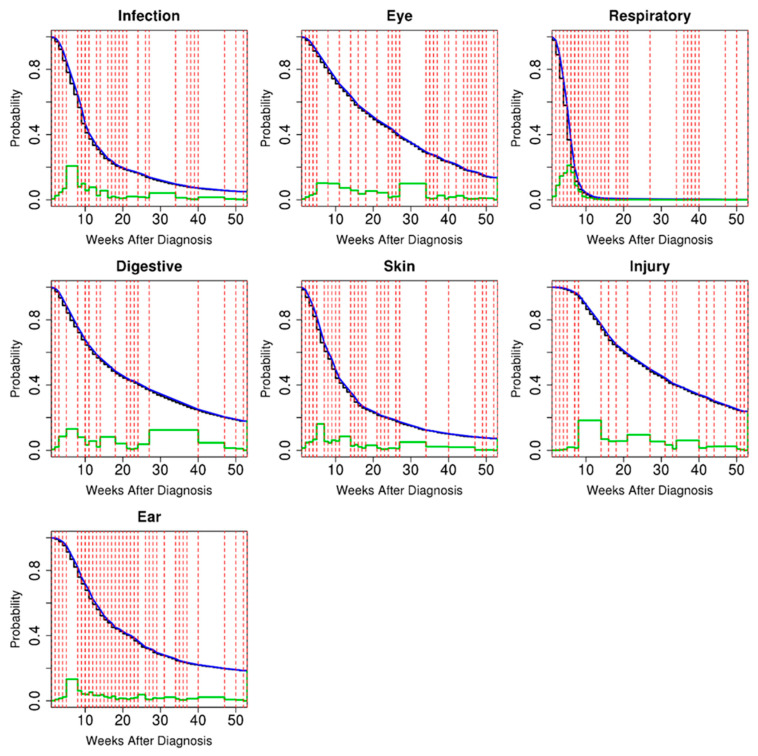
Univariate analyses of single events for the 9533 patients. The probability of the event occurrence was estimated for each disease diagnosis event by CEPA (blue) and the well-known Kaplan–Meier (black) method. The dashed red vertical lines represent the CEPA partitions within which the probability density was expected to be uniform. The green lines represent the probability masses assigned to the CEPA partitions.

**Figure 4 healthcare-12-00939-f004:**
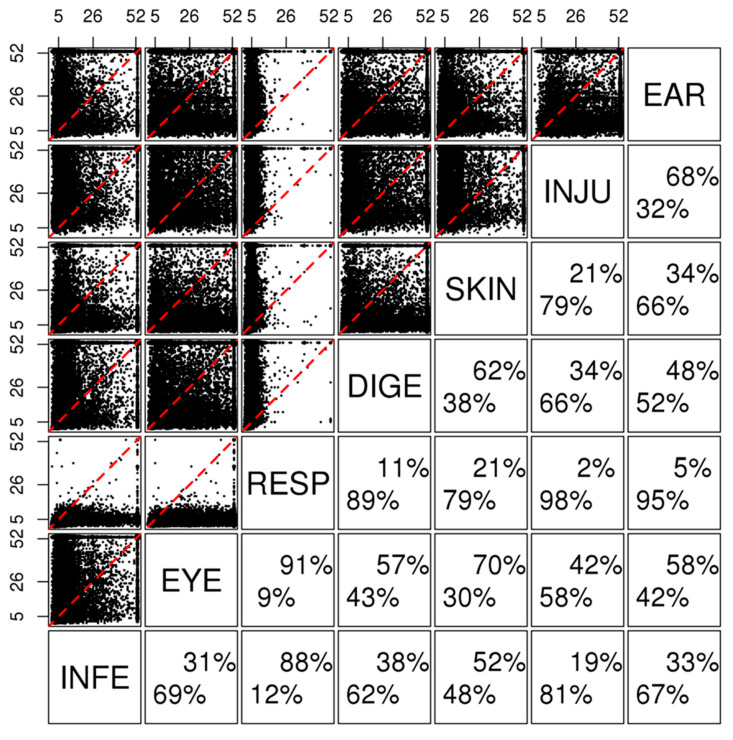
Pairwise precedence of the disease diagnosis events for the 9533 patients. The joint distribution of bivariate time-to-events was estimated by CEPA for each pair of events. The considered events were the diagnoses of infectious disease (INFE), eye disease (EYE), respiratory disease (RESP), digestive disease (DIGE), skin disease (SKIN), injury (INJU), and ear disease (EAR). Each top-left panel shows the observed or censored days of the bottom and right-side events. Each bottom-right panel shows the precedence chance of the top-side events to the left-side events (top-right %) and the opposite case (bottom-left %) when both events occurred at different times. For example, respiratory diseases preceded digestive diseases and were diagnosed 89% of the time.

**Figure 5 healthcare-12-00939-f005:**
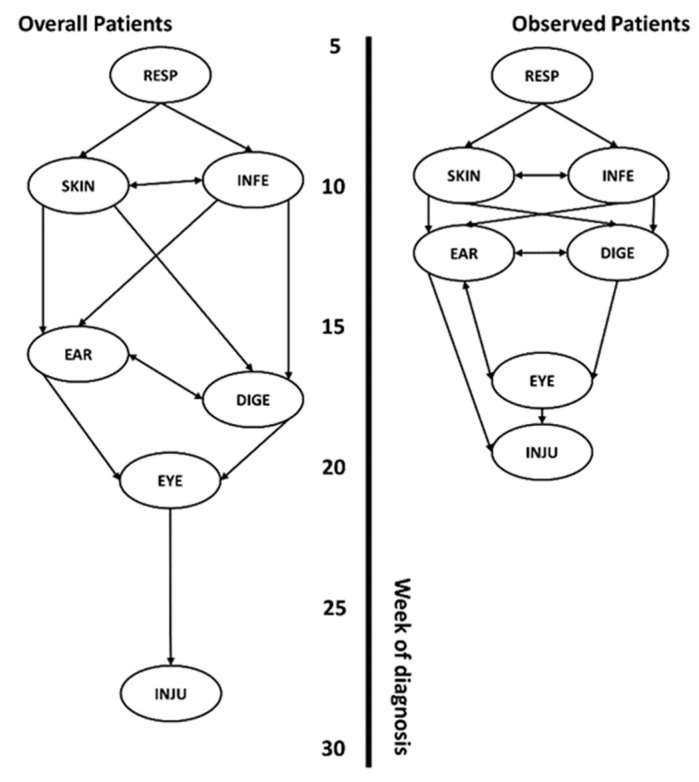
Predicted order of diagnosis of a disease based on valid sequence scores. The left and right panels show the sequences of the overall and observed patients, respectively. RESP, SKIN, INFE, EAR, DIGE, EYE, and INJU represent the disease diagnoses of respiratory, skin, infectious, ear, digestive, eye, and injury diseases, respectively.

**Figure 6 healthcare-12-00939-f006:**
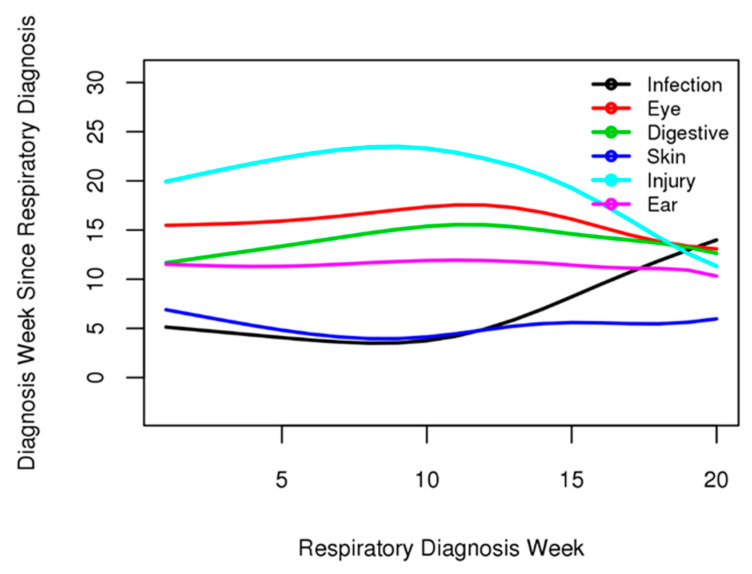
Relationship between the dependence of time on disease diagnosis. Median time to diagnose infectious, eye, digestive, skin, ear, and injury diseases after respiratory disease diagnosis. Digestive, skin, and ear diseases were diagnosed at a certain time after diagnosing respiratory diseases. However, there was no relationship between infectious, eye, and injury diseases.

**Table 1 healthcare-12-00939-t001:** This table presents the results for the seven disease groups with the highest diagnosis rates categorized based on the ICD-10 classification. For ease of analysis, the names of the disease groups are assigned based on the descriptions in the ICD-10, and the table displays the number of diagnosed cases and diagnosis rates for each disease group.

	ICD-10						
	A00-B99	H00-H59	H60-H95	J00-J99	K00-K93	L00-L99	S00-T98
Disease	Infection	Eye	Ear	Respiratory	Digestive	Skin	Injury
Number of patients	9053	8326	7767	9521	7824	8843	7261
Diagnosis rate (%)	94.96	86.39	81.47	99.87	82.07	92.76	76.17

**Table 2 healthcare-12-00939-t002:** Sample distributions followed when generating the time-to-event (T) and censored time (C) of the simulation data. These were generated to satisfy the three conditions for establishing a dependency between two events. $N\left(\mu ,\Sigma \right)$ is a trivariate normal distribution where μ is the mean and Σ is the covariance. $S\left(\cdot \right)$ represents the bivariate survival function, i.e., S($t_{1},\;t_{2})=\mathrm{Pr}[T_{1}>t_{1},T_{2}>t_{2}]$.

Distribution	T	C
Uniform	$T_{1}\sim Unif\left(1\right)$ $T_{2}\sim T_{1}+Z_{1}$ $T_{3}\sim T_{2}+Z_{2}$ $Z_{n}\sim Unif\left(1\right),\;\;\;\;Z_{n}\;⊥\;T_{N}$	$C_{1}\sim Unif\left(1\right)$ $C_{2}\sim Unif\left(2\right)$ $C_{3}\sim Unif\left(3\right)$ $C_{1}\;⊥\;C_{2}$, $C_{2}\;⊥\;C_{3}$, $C_{1}\;⊥\;C_{3}$
Log-normal	$log\left(\begin{array}{c} T_{1}\\ T_{2}\\ T_{3} \end{array}\right)\sim N\left(\left(\begin{array}{c} 0\\ 0.5\\ 1 \end{array}\right),\left(\begin{array}{ccc} 1 & 0.5 & 0.5\\ 0.5 & 1 & 0.5\\ 0.5 & 0.5 & 1 \end{array}\right)\right)$	$log\left(\begin{array}{c} C_{1}\\ C_{2}\\ C_{3} \end{array}\right)\sim N\left(\left(\begin{array}{c} 0\\ 0\\ 0 \end{array}\right),\left(\begin{array}{ccc} 1 & 0 & 0\\ 0 & 1 & 0\\ 0 & 0 & 1 \end{array}\right)\right)$
Additive Exponential	$T_{1}\sim Exp\left(1\right)$ $T_{2}\sim T_{1}+Z_{1}$ $T_{3}\sim T_{2}+Z_{2}$ $Z_{n}\sim Exp\left(1\right),\;\;\;\;Z_{n}\;⊥\;T_{N}$	$C_{1},\;C_{2},C_{3}\;\sim \;Exp\left(0.5\right)$ $C_{1}\;⊥\;C_{2}$, $C_{2}\;⊥\;C_{3}$, $C_{1}\;⊥\;C_{3}$
Clayton	$T_{1}$, $T_{2}$~S($t_{1},\;t_{2})$ = ${\left\{e^{t_{1}/\theta }+e^{t_{2}/\theta }-1\right\}}^{-\theta }$, where θ=1 $T_{3}\sim \;T_{2}+Exp\left(1\right)$	$C_{1},\;C_{2},C_{3}\;\sim \;Exp\left(0.5\right)$ $C_{1}\;⊥\;C_{2}$, $C_{2}\;⊥\;C_{3}$, $C_{1}\;⊥\;C_{3}$

**Table 3 healthcare-12-00939-t003:** Event summary for the disease diagnosis events. Median days of event occurrences were estimated from all the events, including censored events (overall), as well as those from the observed events (observed).

Event	Censoring Rate	Overall Median Day	Observed Median Day
Infection	5.04%	9.7	9.4
Ear	18.5%	15.9	13
Eye	13.61%	20.5	16.7
Respiratory	0.126%	5.4	5.4
Digestive	17.9%	17.3	13
Skin	7.24%	10	9.4
Injury	23.8%	26.9	19.6

**Table 4 healthcare-12-00939-t004:** Top six frequent sequences of events based on the precedence and sequence analysis score function for the 9533 patients. The most probable sequence order with the lowest score was respiratory, skin, infectious, digestive, ear, eye, and injury diseases. Sequences with scores no more than 2% higher than the best score were above the borderline as valid sequences.

Event Sequence	Score	Rate
Respiratory → Skin → Infectious → Digestive → Ear → Eye → Injury	6.90	0.00
Respiratory → Infectious → Skin → Digestive → Ear → Eye → Injury	6.93	0.50
Respiratory → Skin → Infectious → Ear → Digestive → Eye → Injury	6.93	0.50
Respiratory → Infectious → Skin → Ear → Digestive → Eye → Injury	6.97	1.01
Respiratory → Skin → Infectious → Digestive → Eye → Ear → Injury	7.04	2.03
Respiratory → Skin → Infectious → Digestive → Ear → Injury → Eye	7.04	2.03

## Data Availability

The datasets generated and/or analyzed during the current study are available from the National Health Insurance Service (NHIS), which released the National Health Insurance Service—Medical Treatment Details Information dataset in 2016 under an ‘Open with no restriction on use’ license. These data are managed by the Big Data Strategy Department and are available for free download from the NHIS website (https://www.data.go.kr/en/data/15007115/fileData.do, accessed on 10 March 2023). The source codes for this work are publicly available at https://github.com/Jangwon37/CEPA, accessed on 10 March 2023.

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
