# Peer review of "Prioritizing Disease Diagnosis in Neonatal Cohorts through Multivariate Survival Analysis: A Nonparametric Bayesian Approach"

_healthcare, 2024, doi:10.3390/healthcare12090939_

Round 1

Reviewer 1 Report (Previous Reviewer 1)

Comments and Suggestions for Authors

I reread with interest the manuscript entitled "Prioritizing Disease Diagnosis in Neonatal Cohorts Through Multivariate Survival Analysis: A Nonparametric Bayesian Approach"

Although the abstract and introduction have been significantly revised, I ask that the abstract present your entire manuscript more clearly. Again in this form it leaves an overly general impression.

I repeat, you must clearly state what the aim of the study was and what hypothesis you tested.

The materials and methods and results section now meets my expectations.

The discussion is still insufficient. Refocus on previous comments. The discussion must comprehensively elaborate the issue, including clinical and statistical implications, with a clear focus on what we get from your results in relation to previous studies on the mentioned topic, and how they can be applied in the future in the context of similar research. The number of references is insufficient. There are many more relevant articles closely related to this topic.

The conclusion must be more concise, only three to four sentences. Add the rest to the discussion section.

In general, you have improved the manuscript considerably, but the discussion section needs to be far more comprehensive with far more relevant references to the given topic.

Comments on the Quality of English Language

Moderate editing of English language required.

Author Response

Reviewer 2 Report (Previous Reviewer 3)

Comments and Suggestions for Authors

The article is generally good, with a few minor changes listed below.

-The contributions of the article should be presented as bullet points in the introduction section.

-The boxplot chart is presented in Figure 2, but no technical information about the boxplot is shared. Although the middle line represents the median by default, it is sometimes presented as the mean value. This information should be detailed. You can use "10.1016/j.vetimm.2022.110470" boxplot definition is given in this article.

-The image quality of all figures except Figure 1 is good. There is some blur in Figure 1, and it would be better if you color the figure.

Round 2

Reviewer 1 Report (Previous Reviewer 1)

Comments and Suggestions for Authors

Thanks for the answers and improving the manuscript.

Comments on the Quality of English Language

Moderate editing of English language required.

This manuscript is a resubmission of an earlier submission. The following is a list of the peer review reports and author responses from that submission.

Round 1

Reviewer 1 Report

Comments and Suggestions for Authors

I read with interest the manuscript entitled "Prioritizing Disease Diagnosis in Neonatal Cohorts Through Multivariate Survival Analysis: A Nonparametric Bayesian Approach"

The introduction is written too generally with a lot of commonly known information, without a clearly defined aim and hypothesis. The introduction must be concise with a brief overview of the previous knowledge on the mentioned topic, as well as a clearly and concisely set aim of the research.

Also, in the introduction, the main focus is on statistics. Please also include clinical knowledge about the topic you are researching.

Why did you decide on a period of one year (2002)?

What methodology was used to reclassify the disease? Who conducted it? You must clearly specify all 27 diseases in your manuscript and flowchart.

Based on what methodology were the seven disease categories extracted? Please explain.

How did you define the complications of the disease?

Please put a reference to the National Health Insurance Service database, archived in the repository of the Ministry of the Interior and Safety, Republic of Korea.

Within the results section, many sections are about methodology. I ask that you describe all methodological approaches in section 2, while in the results you only list the results.

Observing the results, it is clear that they came from a rough division into 7 variables. Every clinician would wonder what to do with the given data? I repeat, you must clearly state what the goal of the study was and what hypothesis you tested. The use of statistical tools is not disputed, but what did we get from your data and how can we use them to improve the provision of health care?

The discussion should start with a brief presentation of own results, which should then be compared with studies of interest. The focus must be on clinical implications, not a discussion of statistical models. The aforementioned statistical models have already been discussed a lot.

Nothing substantial was discussed within the discussion. Also, it is unacceptable that you do not have a single reference within the discussion. The conclusion does not refer to the essence of the study, but generalizes the issue.

Comments on the Quality of English Language

Moderate editing of English language required.

Reviewer 2 Report

Comments and Suggestions for Authors

After reviewing the Manuscript "Prioritizing Disease Diagnosis in Neonatal Cohorts Through Multivariate Survival Analysis: A Nonparametric Bayesian Approach" The assessor has the following comments:

1. The manuscript presents the method of a non -parameter Bayesian method to directly estimate the density function of the multi -variable survival time. Test results with simulation data include 500 Samples and 9533 Patient with 7 variables from the National Health Insurance Cohort in Korea. Manuscript presented detailed and complete research content as well as test results.

2. The reviewer has only 1 small comment about the Discussion section. The author uses the data of 9565 patient with 7 out of 27 variables. Therefore, add more one paragraph or some sentences to discuss the influence of the remaining 20 variables in this database.

Comments on the Quality of English Language

-

Reviewer 3 Report

Comments and Suggestions for Authors

The article is not a novel.

A limited number of models were used.

Since the codes of the methods were not shared in the study, the transparency of the results is a matter of debate.